# Elastic energy savings and active energy cost in a simple model of running

**Ryan T. Schroeder**[1]*, **Arthur D. Kuo**[1,2]

**1** Faculty of Kinesiology, University of Calgary, Alberta, Canada, **2** Biomedical Engineering Program, University of Calgary, Alberta, Canada

* ryan.schroeder@ucalgary.ca

**Data Availability Statement:** The source code and data used to produce the results and analyses presented in this manuscript are available from the Bitbucket Git repository: https://bitbucket.org/hbcl/runoptsol/src/main/.

## Abstract

The energetic economy of running benefits from tendon and other tissues that store and return elastic energy, thus saving muscles from costly mechanical work. The classic "Spring-mass" computational model successfully explains the forces, displacements and mechanical power of running, as the outcome of dynamical interactions between the body center of mass and a purely elastic spring for the leg. However, the Spring-mass model does not include active muscles and cannot explain the metabolic energy cost of running, whether on level ground or on a slope. Here we add explicit actuation and dissipation to the Spring-mass model, and show how they explain substantial active (and thus costly) work during human running, and much of the associated energetic cost. Dissipation is modeled as modest energy losses (5% of total mechanical energy for running at 3 m s⁻¹) from hysteresis and foot-ground collisions, that must be restored by active work each step. Even with substantial elastic energy return (59% of positive work, comparable to empirical observations), the active work could account for most of the metabolic cost of human running (about 68%, assuming human-like muscle efficiency). We also introduce a previously unappreciated energetic cost for rapid production of force, that helps explain the relatively smooth ground reaction forces of running, and why muscles might also actively perform negative work. With both work and rapid force costs, the model reproduces the energetics of human running at a range of speeds on level ground and on slopes. Although elastic return is key to energy savings, there are still losses that require restorative muscle work, which can cost substantial energy during running.

## Author summary

Running is an energetically economical gait whereby the legs bounce like pogo sticks. Leg tendons act elastically to store and return energy to the body, thus saving the muscles from costly work with each running step. Although elasticity is known to save energy, it does not explain why running still requires considerable effort, and why the muscles still do substantial work. We use a simple computational model to demonstrate two possible reasons why. One is that small amounts of energy are lost when the leg collides with the ground and when the tendons are stretched, and muscles must restore that energy during

**Funding:** This work was funded by the Dr. Benno Nigg Research Chair (ADK) and the National Sciences and Engineering Research Council of Canada (https://www.nserc-crsng.gc.ca/; NSERC CRC, Tier 1 to ADK; NSERC Discovery to ADK). The funders had no role in study design, data collection and analysis, decision to publish, or preparation of the manuscript.

**Competing interests:** The authors have declared that no competing interests exist.

steady running. A second reason is that muscles may perform work to avoid turning on and off rapidly, which may be even more energetically costly. The resulting muscle work, while small in quantity, may still explain most of the energetic cost of running. Economy may be gained from elasticity, but running nonetheless requires muscles to do active work.

## Introduction

Running is distinguished by the spring-like, energy-saving behavior of the stance limb [1–4], analogous to a pogo stick (Fig 1). It is modeled well by a classic analogy, the *Spring-mass model*, where the limb acts elastically to support and redirect the body center of mass (CoM) between flight phases, and all mechanical energy is conserved throughout each step. This simple model can reproduce the motion and forces of running remarkably well and explains how series elastic tissues such as tendon can improve running economy. It applies to bipeds and even polypeds, making it one of the most universal and elegant models for running. However, it does not include muscles that actively contract against series elasticity, and it fails to explain

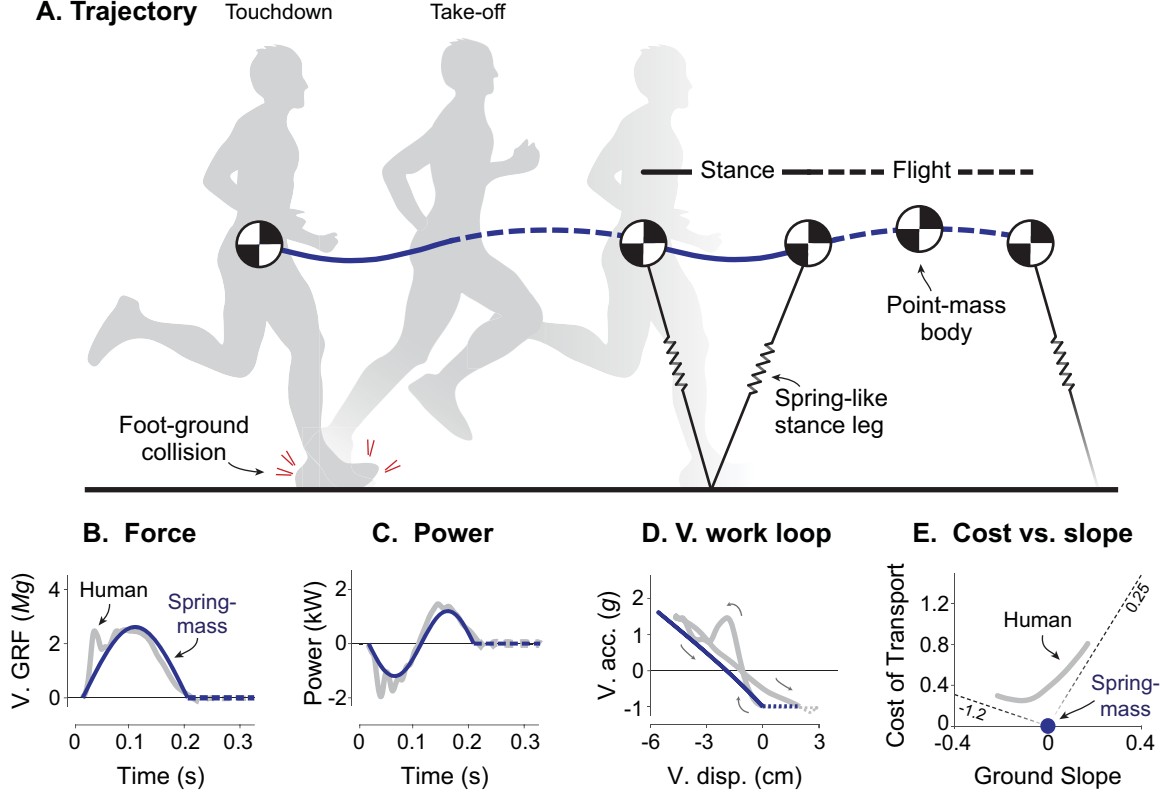

**Fig 1. Human running and the Spring-mass model.** (A) Human stance phases resemble motion of a spring-mass system with no energy loss, alternating with parabolic Flight phases. Body mass is lumped into a single point center of mass (CoM). Traces of (B) vertical ground reaction forces (V. GRF) vs. Time, (C) leg Power vs. Time, (D) and vertical acceleration vs. displacement (V. acc. vs. V. disp.; termed *vertical work loop curve*) are all shown for both human data (gray lines) and the Spring-mass model (dark solid lines). (E) The energetic Cost of Transport (cost per unit weight and distance) for humans running on slopes (after [11]) is not explained by the Spring-mass model, which only operates at zero ground slope and zero energetic cost. The spring-like behaviors (B-D) should be regarded as pseudo-elastic, because humans and other animals experience dissipation such as in foot-ground collision, and thus, require active muscle actuation.

 

the substantial metabolic cost measured during running. An extension of the Spring-mass model to include active actuation may help explain organismal running energetics and how best to exploit series elasticity for economy.

The Spring-mass model agrees well with a wide body of experimental evidence [4]. It reproduces mechanical characteristics such as the body's trajectory in space (Fig 1A), ground reaction forces (Fig 1B), leg mechanical power (Fig 1C), and even the leg's vertical work loop curve (vertical acceleration vs. vertical displacement similar to an elastic spring's work loop, Fig 1E; [4]). The spring can passively store and return mechanical energy to the CoM, reducing the active work otherwise required of active muscle, and thus, improve running economy. Some have therefore proposed that more compliance or longer tendons are key to running economy [5,6]. For example, the energetic cost of human running is less than half of that expected if muscles alone performed work on the CoM [7]. In turkeys, tendon contributes over 60% of the shortening work performed by the lateral gastrocnemius [8]. Although the simple Spring-mass model (Fig 2A) applies mainly to bipedal running or polypedal trotting, multiple leg springs can reproduce galloping, and indeed, practically all of the running gaits observed in nature [9,10]. Few other models reproduce so many behaviors with such simplicity.

There are also important aspects of running not captured by the Spring-mass model. A critical feature is metabolic energy expenditure by muscle [12], considered important for selecting gait and speed [13,14], and more generally for a variety of animal behaviors [15]. The Spring-mass model is conservative of mechanical energy and predicts no such expenditure. Lacking muscle actuation, it is incapable of accelerating from rest or running on sloped ground. Steeper slopes in particular have energetic costs approaching that expected from muscles performing positive and negative work against gravity at their respective efficiencies (Fig 1E; [11]). Even steady running on the level entails substantial muscle shortening work, as shown in turkeys (e.g., 40% of muscle-tendon work; [8]), and in human running [16,17]. Some of that work is fundamentally necessary because of dissipation, for example by tendons with hysteresis (26% loss per cycle in Achilles tendon during hopping; [18]), by the heel pad [19] and other soft tissues that deform (33% per step of human running at 3 m s$^{-1}$, [20]). Restoration of those losses alone could account for up to 29% of the energetic cost of human running [20], and the overall active work of muscle for as much as 76% of the energetic cost of human running [21]. The spring-like mechanics of running (Fig 1B–1D) should therefore be regarded as pseudo-elastic, as opposed to purely elastic. Beyond the conceptual illustration of energy savings, the Spring-mass does not account for dissipation and is not predictive of actual energy costs observed in nature.

Other simple models of running have included elements other than springs. Perhaps the simplest of these has only an active, extending actuator (Actuator-only model, Fig 2B; [14]). Minimization of its work alone is sufficient for both walking and running to emerge as optimal gaits, with running more economical at faster speeds and exhibiting a pseudo-elastic, bouncing-like stance phase despite no passive elasticity [14]. In addition to mechanical work as a cost, we have proposed that muscles also expend energy for a *force-rate cost*, associated with rapid production of force [22–25], which helps to explain human-like ground reaction forces [26]. Others have optimized the actuation of robots, including both elasticity and dissipative elements, and have shown a variety of running gaits to emerge [27,28]. For organisms, similar models have been used to explore stability [29] and economical strategies [30] of terrain navigation. Still, while dissipation has been characterized empirically (e.g., [31]), most running energetics models have not included such dissipation with series elasticity and actuation to explain energy cost.

Here we propose a running model that combines series elasticity with active actuation and passive dissipation (Fig 2). The classic Spring-mass (Fig 2A) and pure Actuator-only (Fig 2B)

 

### A. Spring-mass          B. Actuator-only          C.  Actuated Spring-mass

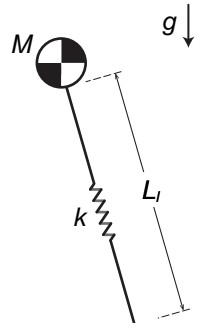 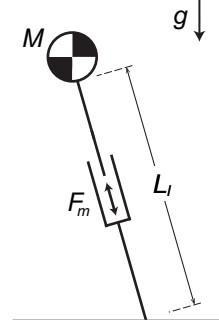 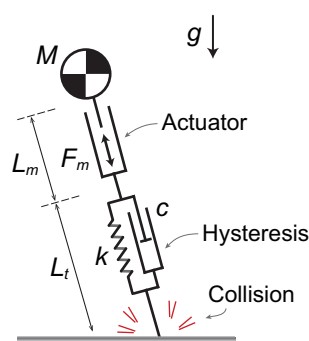

**Fig 2. Simple running models with and without elastic spring, active actuator, and dissipative elements.** (A) The Spring-mass model comprises a point-mass body and a massless spring for a leg. (B) The Actuator-only model replaces the spring with a massless, active actuator producing extension forces in the leg [14]. (C) The proposed Actuated Spring-mass model combines an actuator and a spring (analogous to a muscle-tendon unit), along with two passive, dissipative elements: a damper in parallel with the spring to model tendon hysteresis, and collision loss to model dissipation of kinetic energy at touchdown. In the models, $g$ is gravitational acceleration and $M$ is body mass. $L_l(t)$, $L_t(t)$, and $L_m(t)$ are time-varying lengths of the leg, spring and actuator, respectively. Parameters $k$ and $c$ are spring stiffness and damping coefficient. $F_m(t)$ is the active actuator force in the leg's extension direction.

models serve as opposing reference points that can produce running mechanics (Fig 1A–1D) with and without elasticity. We propose to combine the spring and actuator in series, along with dissipation, in an *Actuated Spring-mass model* (Fig 2C) that may be more representative of running in organisms. We expect that such a model will leverage series elasticity to perform minimal work, as needed to restore dissipative losses. We test whether such a model is sufficient to explain both the mechanics and metabolic cost of running (Fig 1E), with work minimization as the sole objective, or work plus the proposed force-rate cost. Such a model may help determine whether more compliant tendon is indeed economical [6], and provide insight on the energy expended by muscle. We use human data for running at different speeds and ground slopes as experimental comparison, but the principles revealed by the model are intended to help explain running across a range of animal species.

## Methods

We used dynamic optimization to determine optimal actuation strategies for the proposed Actuator-Spring-mass running model. The model extends the classic Spring-mass model by adding an active series actuator and dissipative losses (Fig 2C). The actuator can perform positive and/or negative work, in part to compensate for two modes of passive energy dissipation: collision loss associated with foot-ground contact and hysteresis of the tendon spring. The model was optimized for energy economy, as defined by the energetic costs of that active actuator work, briefly summarized here.

Two of the most basic elements of the model are the mass and elastic spring. The point mass $M$ was supported by a spring with stiffness $k$, which was varied as a free parameter to produce a wide spectrum of gaits. These included low stiffnesses ranging from grounded running with no flight phase, to more impulsive running with a brief stance period and a relatively long flight phase. The limiting case of impulsive running has infinitesimal stance and an infinitely stiff spring, where the body's motion is almost entirely described by its parabolic trajectory during flight.

Two types of dissipation were included in the model, representing losses from collisions and hysteresis. Collisions model the kinetic energy dissipated when the body impacts the ground. For example, humans lose momentum associated with 2.6–7.8% body mass [31]. We modeled this as a simple discontinuity in the CoM velocity vector magnitude at touchdown, defining *collision fraction* (CF) as the fraction of momentum lost in the collision. A nominal collision fraction of 3% resulted in a 5.9% loss in kinetic energy (see S1 Text for details).

Hysteresis was included to model the imperfect energy return of tendons and other series elastic tissues. Hysteretic energy losses of 10–35% per stretch-shortening cycle have been estimated *in vivo* for tissues such as the human Achilles tendon [32]. Estimates of soft tissue deformation suggest that much of the actual dissipation occurs during the first half of stance [20], modeled with a viscous damper (in parallel with the spring) only dissipating energy during spring loading (Fig 2C). Damping was parameterized by damping ratio, $\zeta = c/\sqrt{4Mk}$, where $c$ is the damping coefficient. A nominal damping value of $\zeta = 0.1$ was selected to yield 26% hysteresis, roughly within the estimated range of humans.

Each stance phase was computed with dynamic optimization for energy economy. The main control variable was the time-varying actuator length during the leg's stance (specifically, its third derivative $\dddot{L}_m(t)$ was used to allow for calculations of force rate during implementation), treating the stance and swing phases as periodic and symmetric between legs. The objective function to be minimized was the energy cost per step $E$,

$$E = E_W + E_R, \tag{1}$$

as the sum of a cost $E_W$ for work, and another cost $E_R$ for force rate. The work cost depends on positive and negative efficiencies for muscle, $\eta^+$ and $\eta^-$ respectively (25% and -120% from [11]), defined as work divided by metabolic energy (superscript + or − for positive and negative work, respectively). The energetic cost was therefore defined by actuator work ($W_m$ per step, and power $P_m$ for work per time, superscripts for positive and negative work),

$$E_W = \int_0^T \left( \frac{P_m^+}{\eta^+} - \frac{P_m^-}{\eta^-} \right) dt \tag{2}$$

Force-rate was added as a separate cost from work, motivated by two observations. First, metabolic cost has been observed to increase with intermittent bursts or rapid cycles of force, even under isometric conditions where little or no work is performed [24,33–35], or in cyclic movements where work is kept constant [23,25]. Second, Actuator-only models with work as the only cost result in unrealistically impulsive ground reaction forces [14], whereas the addition of a force-rate cost produces more human-like forces [26]. We therefore included a force-rate cost $E_R$, increasing with the integral of the force rate squared over a single step,

$$E_R = \varepsilon \int_0^T \dot{F}_m^2 \, dt \tag{3}$$

where $\dot{F}_m$ is the first derivative of actuator force and $\varepsilon$ is a cost coefficient that converts the mechanical quantity into units of energy. Actuator force and force rate were determined from the spring-damper force and its derivative, respectively (see S1 Text for details). Variations on this formulation of force-rate cost have been examined in previous locomotion models, with differences in the exponent or degree of derivative [26,36–41]. But these variations often produce relatively similar effects on optimal gait models [26], and so a single representative formulation is used here.

The optimization was subject to boundary conditions for periodicity and continuity with a ballistic flight phase. Additional constraints included a maximum allowable stance leg length $L$, and vertical ground reaction forces only able to act upward. The optimization determined

the leg's posture at touchdown and did not allow slipping of that contact. Model states included the position ($x_b$, $y_b$) and velocity ($\dot{x}_b$, $\dot{y}_b$) of the point-mass body and the first and second derivatives of the actuator's length ($\dot{L}_m$, $\ddot{L}_m$) to facilitate inclusion of work and force-rate in the objective function.

All optimizations were conducted with the MATLAB software GPOPS-II [42] and the resulting nonlinear problem was solved using SNOPT [43]. All variables and equations were non-dimensionalized with parameter combinations ($L$ = 0.90 m, $M$ = 70 kg and $g$ = 9.81 m s$^2$) during optimization and outputs were subsequently re-dimensionalized as indicated in figures. Further details regarding the model and implementation of the optimization problem can be found in S1 Text.

Running parameters were chosen to represent human-like gait. Speed $v$ was varied over a range of 2–4 m s$^{-1}$, with empirical preferred step frequency given by $f$ = 0.26$v$ + 2.17 [44]. Step length $s$ was defined as the distance travelled over one periodic step, and step frequency $f$ as the inverse of time duration, $T$, per step (i.e. $T$ = 1/$f$), such that $v$ = $s\,f$. Gaits were produced while varying parameters such as tendon stiffness $k$ (4.93–122 kN m$^{-1}$ or equivalently, 6.46–160 $MgL^{-1}$) and force-rate coefficient $\varepsilon$ (0–2·10$^{-2}$ $M^{-1}g^{-1.5}L^{1.5}$). Furthermore, a single set of nominal parameter values (force-rate coefficient $\varepsilon$, spring stiffness $k$ and negative and positive work efficiency $\eta^-$ and $\eta^+$) were selected for comparisons with human ground reaction forces and metabolic data (see Table 1).

For comparison with our model, we included representative human data to qualitatively illustrate well-established patterns for ground reaction forces and other trajectories. The data consist of one representative subject from a separate published study [45]: a male (25 years, body mass = 75.3 kg, leg length = 0.79 m) running on an instrumented force treadmill at 3.9 m s$^{-1}$. An average step was determined from 20 s of steady-state ground reaction force data and used in plots including CoM power [46] and vertical acceleration vs. displacement [2,4], as comparison against the model. These plots reproduce patterns from accepted literature, and so no statistical analysis was performed.

## Results

Optimization results are presented in two parts, first examining the effects of individual model components (Fig 2), and then combining them into a single, unified model. Part I presents

**Table 1. Model parameters and values.**

| Symbol | Description | Range | Nominal | Units |
|---|---|---|---|---|
| $\eta$- | negative work efficiency | | -1.05 | |
| $\eta^+$ | positive work efficiency | | 0.32 | |
| CF | collision fraction | 0–0.06 | 0.03 | |
| $\zeta$ | damping ratio | 0–0.2 | 0.1 | |
| $k$ | spring stiffness | 6.46–160 | 46.7 | $MgL^{-1}$ |
| $\varepsilon$ | force-rate coefficient | 0–2·10$^{-2}$ | 5·10$^{-4}$ | $M^{-1}g^{-1.5}L^{1.5}$ |
| $v$ | running speed | 0.67–1.35 | 1.01 | $g^{0.5}L^{0.5}$ |
| $M$ | body mass | | 70 | kg |
| $g$ | gravitational acceleration | | 9.81 | m s$^{-2}$ |
| $L$ | maximum leg length | | 0.9 | m |

Parameters used in the Actuated Spring-mass model, along with the ranges of values examined for parameter sensitivity analysis, and nominal values for comparison with human. Range is left empty if the parameter was not varied in optimizations. Units are left empty if the parameter has no units. The model was implemented in normalized units, with $M$, $g$ and $L$ as base units (nominal human values shown).

individual sensitivity studies, beginning with the Spring-mass and Actuator-only models, which have been examined in prior literature: e.g. [2,4] and [26,36,40], respectively. Next, the Actuated Spring-mass model is evaluated as an alternative, since it still uses a spring but also requires an actuator to account for passive energy dissipation occurring with each step. Initially, this model is evaluated with the cost of work only (i.e. zero force-rate coefficient $\varepsilon$) over varying speeds and spring stiffnesses. Next, non-zero force-rate coefficients $\varepsilon$ are introduced so changes in actuation strategies may be independently evaluated. Finally, in Part II, a single, unified set of parameters is applied to the Actuated Spring-mass model, which is then used to simulate gait over a range of running speeds, spring stiffnesses and ground slopes to assess its utility in predicting locomotion energetics and optimal spring-actuator coordination patterns.

## Part I: Individual model components and their contributions to running behavior

**Spring-mass and Actuator-only models produce similar pseudo-elastic running gaits.** Both models (Fig 3) can produce similar gaits, ranging from very flat to very bouncy. For the Spring-mass model at a given speed and step length, the spring stiffness determines the gait trajectory, as described by CoM trajectories, vertical ground reaction force profiles, mechanical power profiles, and vertical work loop curves illustrating the spring-like leg behavior [4]. As reported by others [3], the Spring-mass model can run with a vast range of stiffnesses $k$ (Fig 3, top). A less stiff (or more compliant) spring can produce grounded running, in which the stance phase occupies the entire step (blue curves in Fig 3). With greater stiffness comes a flight phase, yielding gaits that resemble more typical human running, where both stance and flight phases are finite in duration (redder curves, Fig 3). These gaits generally include a single-peaked ground reaction force profile, with higher peak forces and powers with increasing

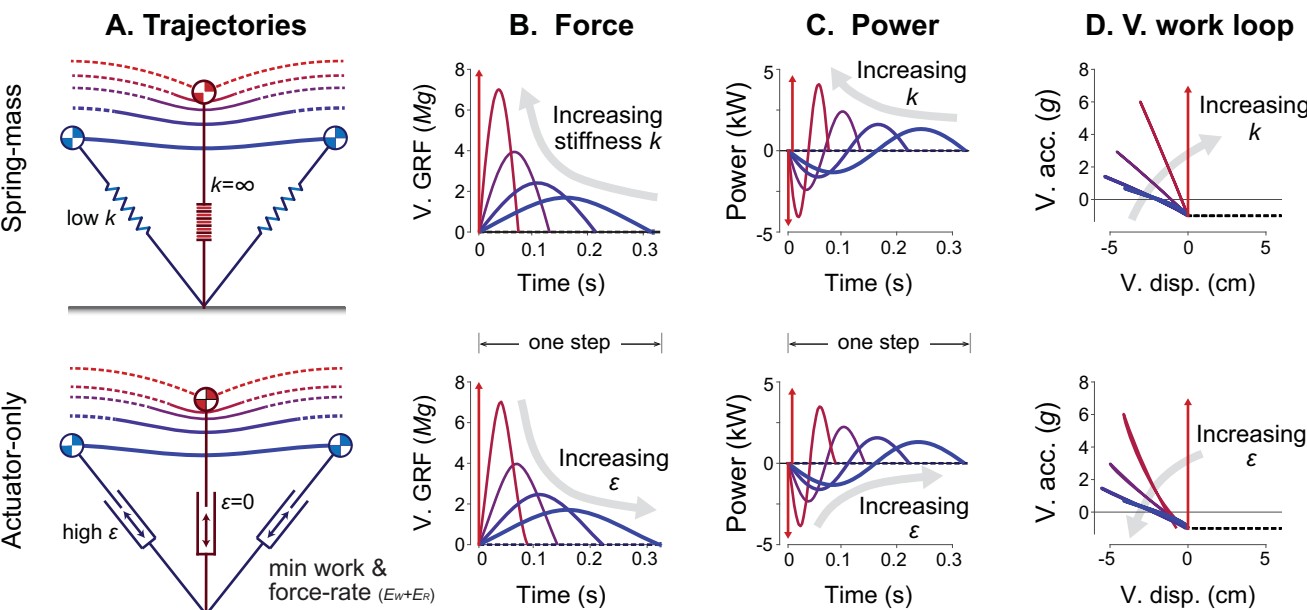

**Fig 3. Running gaits from the Spring-mass (top) and Actuator-only (bottom) models.** They are illustrated by (A) center of mass (CoM) trajectory, (B) vertical ground reaction forces vs. time, (C) leg power performed on the CoM vs. time, and (D) vertical acceleration vs. vertical displacement of the body (or vertical work loop curve). A range of running gaits are shown, varying stiffness $k$ in the Spring-mass model, and the force-rate cost coefficient $\varepsilon$ in the Actuator-only model, for a single running speed $v$ (3.5 m s$^{-1}$) and step frequency $f$ (3 Hz). In the limiting case of infinite spring stiffness or zero force-rate cost, touchdown forces become perfectly impulsive (red arrows).

stiffness, as well as steeper vertical work loop curves during stance. In the limit toward infinite stiffness, the model produces impulsive running [14], where flight takes up nearly the entire step and stance occurs as an instantaneous impulse (red arrows, Fig 3). In all cases, the Spring-mass model is purely elastic and has no actuator and no losses. Thus, no single spring stiffness, and no single spring-like gait, can be considered beneficial over another in terms of energy cost.

The Actuator-only model can produce a very similar range of running gaits, despite the complete lack of an elastic spring (Fig 3, bottom). The optimization produces pseudo-elastic behavior resembling a spring, and tuned by a single parameter: the force-rate coefficient $\varepsilon$. With a coefficient of zero (i.e. work cost only), impulsive running is optimal, because least work is performed with least displacement, albeit with infinite force [14]. A greater force-rate cost results in increasing stance time and shorter flight time, more similar to humans. Increasing that cost further eventually causes the flight phase to disappear, producing grounded running similar to a very compliant spring. For any non-zero force-rate cost, the model consistently produces an approximately linear vertical work loop curve, similar to the Spring-mass model. However, this behavior is purely active and requires substantial positive and negative work. As a result, the Actuator-only model does incur an energy cost for work but has no passive elasticity to reduce that work.

We thus find that the two diametrically opposed models can reproduce the pseudo-elastic behaviors similar to humans, whether or not there is true passive elasticity. There is certainly strong evidence that elasticity is important for running in humans and other animals, but spring-like ground reaction forces and vertical work loop curves are not necessarily indicative that elasticity is the dominant mechanism in running. If it were, the energetic cost of running might be expected to be close to zero. Conversely, the pure Actuator-only model also obviously cannot demonstrate that humans are purely inelastic. The work performed by humans, if there were no elasticity, would result in unrealistically high muscle efficiencies of at least 45% [7]. It is more realistic to regard the human as having some combination of series elasticity and active actuation, both contributing to the actual energetic cost of human running.

## Dissipative energy losses require compensatory, active positive mechanical work

We next consider the effect of passive energy dissipation in the Actuated Spring-mass model (Fig 4), optimizing for the cost of work alone (with zero force-rate coefficient). Again, gaits roughly similar to those of humans are produced, for either stiff or compliant springs. However, the actuator must perform positive work to restore the lost energy. With a stiff spring, it is optimal to produce a relatively "bouncy" CoM trajectory where the body spends more time in the air and thus, reaches greater heights above the ground (Fig 4A). The optimum also favors a spikier vertical force and power over shorter stance durations (Fig 4B and 4C). Conversely, a compliant spring makes it optimal to produce a "flatter" CoM trajectory with briefer flight time (Fig 4A), with lower peak vertical force, longer stance duration, and less stiff vertical work loop curves (Fig 4D). The accompanying leg angle at touchdown also varies, with a more vertical orientation for increasing spring stiffness.

With zero force-rate coefficient $\varepsilon$, it is generally optimal to perform only positive actuator work. For steady motion, the energy lost to hysteresis and collision must be restored with an equal magnitude of positive work, to yield zero net work per step. The optimization reveals this is performed most economically in the second half of stance (Fig 4C, blue shaded areas), in concert with elastic energy return from the spring. It is also generally economical to completely avoid active negative work, which would also require an additional amount of

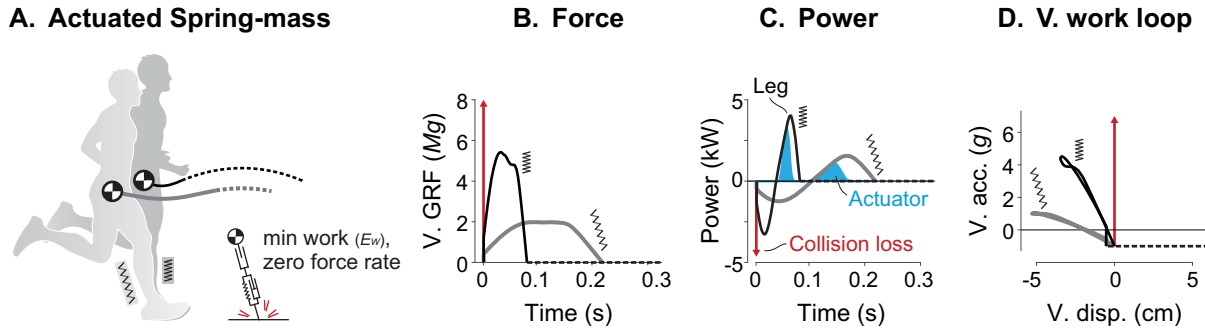

**Fig 4. Effects of stiff vs. compliant springs on Actuated Spring-mass model minimizing cost of work (with zero force-rate cost).** Optimal running gaits (speed $v$ of 3.0 m s$^{-1}$) are shown for the Actuated Spring-mass model, including (A) CoM trajectory, (B) vertical ground reaction forces (V. GRF) vs. time, (C) leg power vs. time, and (D) vertical acceleration vs. vertical displacement (V. acc. vs. V. disp.). The model includes passive dissipation (hysteresis and collision), optimized for two spring stiffnesses $k$ (13.7 kN m$^{-1}$ for compliant and 109.5 kN m$^{-1}$ for stiff). The stiffer spring yields a more vertical leg, shorter stance time and bouncier gait, with higher peak forces and leg power. Net actuator work is similar in both cases.

positive work to be performed. Thus, actuator work is minimized when it is performed only to restore dissipative losses.

## Dissipative losses are minimized by increasing spring stiffness with running speed

We next evaluate a range of spring stiffnesses and running speeds to determine how the cost of actuator work may be minimized (Fig 5). Here we find that optimal spring stiffness increases with running speed (Fig 5A), while generally preserving the timing of positive work within the second half of stance (Fig 5B). At lower speed $v$ (2.5 m s$^{-1}$; Fig 5A left), both hysteresis and collision losses are reduced with less spring stiffness. However, at higher speed (3.5 m s$^{-1}$; Fig 5A right), hysteresis losses are reduced with greater stiffness whereas collision losses are relatively unchanged. For a moderate speed (3.0 m s$^{-1}$; Fig 5A middle) both hysteresis and collision trade off over stiffness, and an intermediate stiffness is optimal. Overall, hysteresis losses change mainly with spring stiffness, and collision losses increase mainly with running speed, so that losses are generally minimized by a stiffness increasing with speed.

The collision and hysteresis losses have distinct dependencies on running speed and/or stiffness. The model's collision loss increases with touchdown velocity and thus, running speed, but is relatively insensitive to spring stiffness. Stiffness does affect the CoM trajectory and the distribution between horizontal and vertical velocity components but has relatively little effect on the vector magnitude. Overall, collision losses increase with speed but are relatively unaffected by spring stiffness (Fig 5).

In contrast, hysteresis loss occurs as a fraction of elastic strain energy, which is largely determined by the angle of the leg during stance. For example, a vertical leg posture is used in conjunction with a stiff spring (Fig 4), and this results in greater strain (and hysteresis losses) to redirect vertical CoM velocity of the bouncier gait. Alternatively, a less vertical leg posture is used with a compliant spring and results in greater strain to redirect horizontal velocity of the body. As such, stiff springs allow for efficient gait at higher speeds, since the vertical leg is effective at mitigating excessive strain to redirect high horizontal CoM velocity. At lower speeds, compliant springs are better since a less vertical leg is better at mitigating higher vertical velocity associated with bouncy running at these speeds.

The overall effects of dissipation are as follows. At low speeds, both collision and hysteresis losses are reduced with relatively low spring stiffness and a shallower leg touchdown angle,

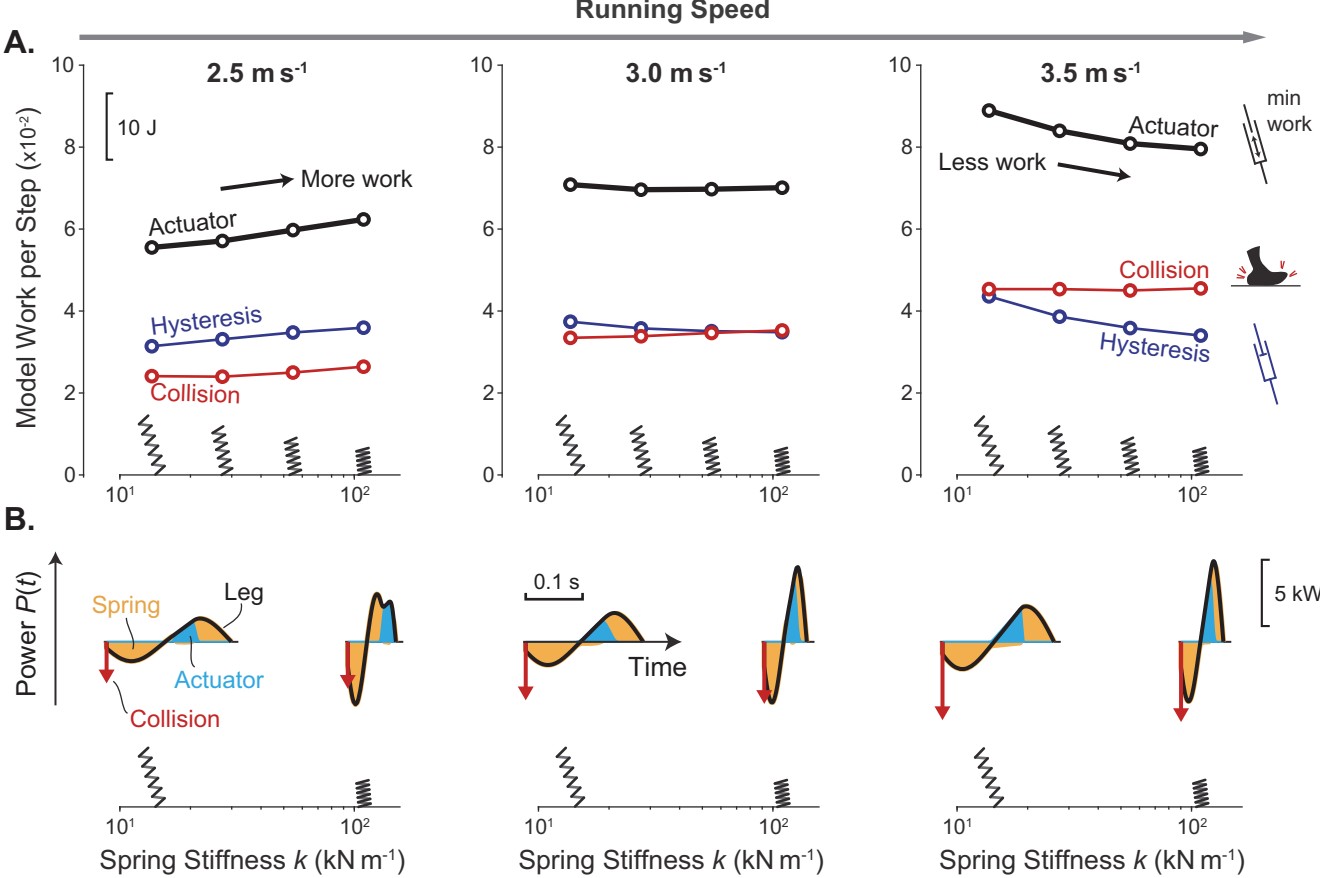

**Fig 5. Actuator work and power as a function of spring stiffness and running speed in the Actuated Spring-mass model, minimizing cost of work (with zero force-rate cost).** (A) Work vs. stiffness for speeds of 2.5–3.5 m s⁻¹. Shown are active Actuator work (black), Collision work magnitude (red), and Hysteresis work magnitude (blue). Spring diagrams (inset) illustrate touchdown angles for each stiffness. (B) Power vs. time for very compliant and very stiff springs, for each running speed. Shown are net Leg power (black lines), Spring power (orange shaded area), Actuator power (blue shaded area). Results are for spring stiffness ranging 13.7–109.5 kN m⁻¹.

thereby also reducing actuator work. But at higher speeds, hysteresis losses are actually reduced by greater spring stiffness and steeper leg touchdown angle, so that actuator work is minimized with relatively high stiffness. These effects together cause the optimal spring stiffness for minimizing actuator work to increase with running speed.

## An added force-rate cost favors active actuator dissipation

We have thus far found that the model with zero force-rate cost avoids active negative work, whereas some animal muscles are observed to perform non-negligible negative work during running [5]. It may seem uneconomical to perform any amount of active negative work, because it only adds to the costly positive work needed to restore the losses. Perhaps there is some indirect energetic advantage to active negative work, not explained by a cost for work alone. In fact, the addition of a force-rate cost with coefficient $\varepsilon$ (Fig 6) makes it favorable for the actuator to perform both negative and positive work. This distributes ground reaction forces over a longer stance duration with lower peak forces and reduced force rate. However, it also comes at the expense of additional actuator work, which is made worthwhile by its capacity to reduce force rate (Fig 6). Overall, the force-rate cost yields less impulsive forces and smoother CoM trajectories, at the expense of active dissipation and increased work.

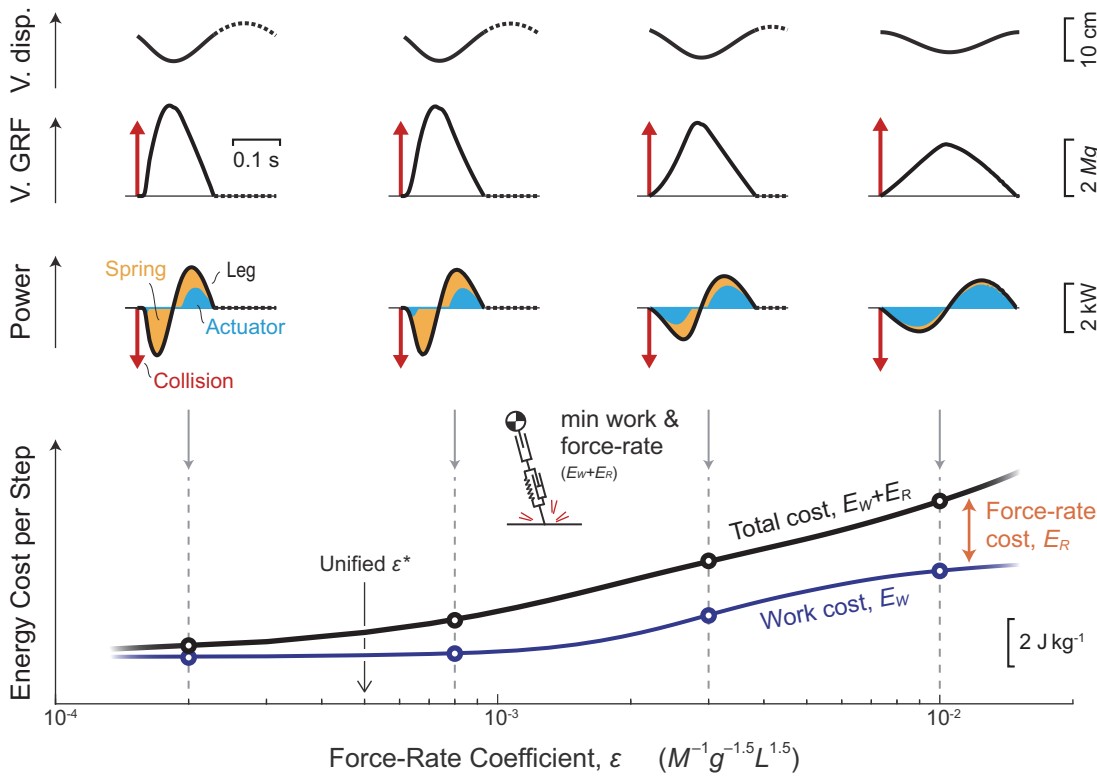

**Fig 6. Effect of work and force-rate costs on running using Actuated Spring-mass model.** (top:) Vertical CoM displacement vs. time, vertical GRF vs. time, and leg Power vs. time, for varying force-rate cost coefficient $\varepsilon$. (bottom:) Actuator work cost $E_W$ (thin blue line), Total cost $E_W + E_R$ (work and force rate, solid black line), and Force-rate cost $E_R$ (difference between lines) vary with the coefficient. Impulsive actions (red arrows, V. GRF and leg Power) occur at Collision, and overall leg power (solid black line) includes contributions from the Spring and Actuator. All solutions are shown for $v$ of 3 m s$^{-1}$ and $f$ of 2.94 Hz.

The added force-rate cost yields a shift in timing of positive work so that it is optimally performed late in stance. Experiments have shown that the triceps surae muscles undergo substantial shortening throughout stance, but particularly late in stance, during human running [16,17]. On the other hand, negative work is optimally performed early in stance as ground reaction forces rise [5,47].

## Part II: Unified actuated spring-mass model of human-like running and energy expenditure

We next apply the Actuated Spring-mass model with a single, unified set of parameter values selected to produce human-like running in a variety of conditions. The full model therefore includes an elastic spring, both hysteresis and collision losses, and an objective to minimize both work and force rate costs. A single force-rate coefficient is selected ($\varepsilon^*$ of $0.5 \cdot 10^{-3}$) to approximately match the model's output to human data, along with stiffness $k^*$ of 35.6 kN m$^{-1}$, positive work efficiency $\eta^+$ of 32%, and negative work efficiency $\eta^-$ of -105%. The resulting model, with parameters thus fixed, is then applied to three comparisons with human data: mechanics of a nominal gait, energetic cost as a function of running speed, and energetic cost as a function of ground slope.

## Unified model produces human-like gait mechanics

The resulting model qualitatively matches the human CoM trajectory (Fig 7A), vertical ground reaction forces (Fig 7B), leg power vs. time (Fig 7C), and vertical acceleration vs. displacement

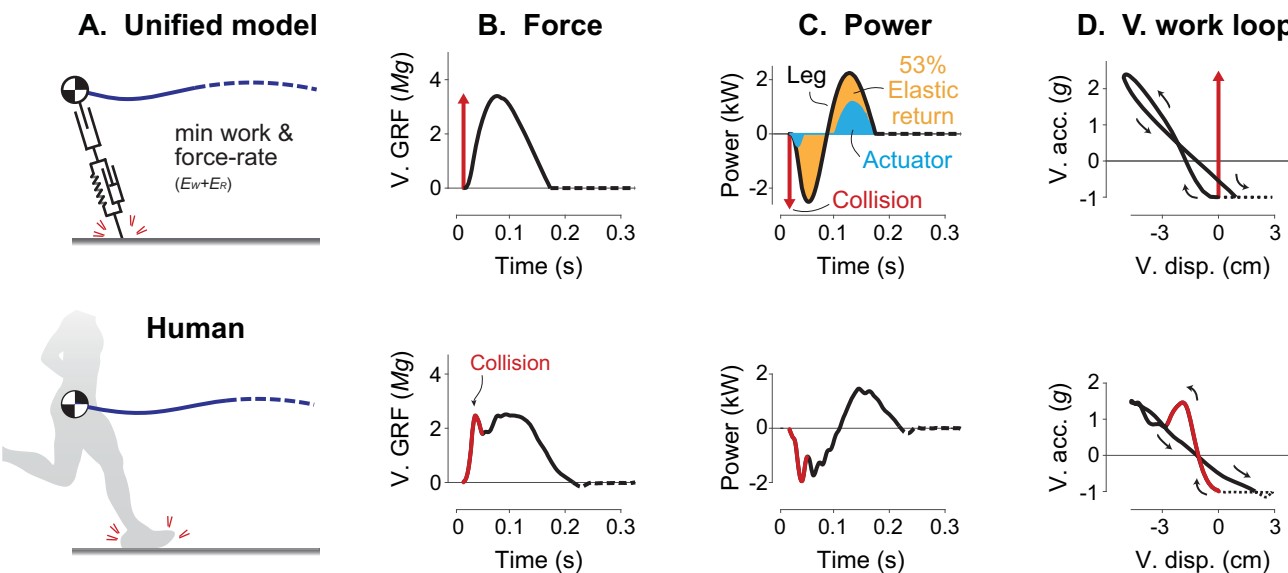

**Fig 7. Comparison of Unified Actuated Spring-mass model (top) including work and force-rate costs against human data (bottom).** Shown are (A) CoM trajectories, (B) vertical ground reaction forces, (C) leg mechanical power, and (D) vertical acceleration vs. displacement. Initial force transients are highlighted (red impulse arrow for model, red line for human). In (C), spring (orange shaded area) and actuator work (blue shaded area) contributions are shown. Gait parameters $v$ and $f$ are 3.9 m s$^{-1}$ and 3 Hz, respectively. Stiffness and force-rate coefficient in the model are selected to approximately match stance time duration: $k^*$ is 35.6 kN m$^{-1}$ and $\varepsilon^*$ is $0.5 \cdot 10^{-3}$.

(Fig 7D). The model also reproduces some features that the classical Spring-mass model cannot, such as the brief initial peak at touchdown (from collision [31], Fig 7B) and a more gradual decrease in force than the increase (temporal asymmetry [30], Fig 7B). The model collision produces a transient burst of negative power [46] followed by elastic energy storage and return (about 53%; Fig 7C), as well as slightly non-linear vertical work loop curves (as in [4], Fig 7D), qualitatively similar to human. Whereas the Spring-mass model produces a nearly linear curve that retraces itself almost perfectly (Fig 1D), the human curve has an initial transient and self-intersecting profile resembling a tilted figure-eight. The shape indicates hysteresis with a dissipative counter-clockwise loop, followed by a (positive work) clockwise loop. The present model crudely reproduces these broad features, even if imperfect in detail.

## Unified model has increasing energy cost with speed for level running

The model's energetic cost per time (Fig 8A) increases with running speed at a rate similar to human data [48]. Here, the model's step frequency was constrained to the empirical human preferred step frequency, but other parameters were kept fixed. The increasing overall cost with speed may be explained by the constituent force-rate and work costs (Fig 8B), evaluated as a function of spring stiffness and speed. The work cost is considerably greater in magnitude than the force-rate cost (e.g., 68% vs. 32%, respectively at 3 m s$^{-1}$) and increases more as a function of speed, primarily for restoring collision losses. Thus, most of the model's overall cost for running at higher speeds is due to increased actuator work, which is not included in the Spring-mass model.

Nevertheless, the force-rate cost has a large influence on the model's gait as a function of spring stiffness (Fig 8B). Greater stiffness is associated with more impulsive ground reaction forces and briefer stance durations (as in Fig 4), thus resulting in higher force-rate cost. Furthermore, the actuator performs additional negative (and therefore also positive) work with greater stiffness as a trade-off against even higher force-rate costs (like in Fig 6). Overall, the

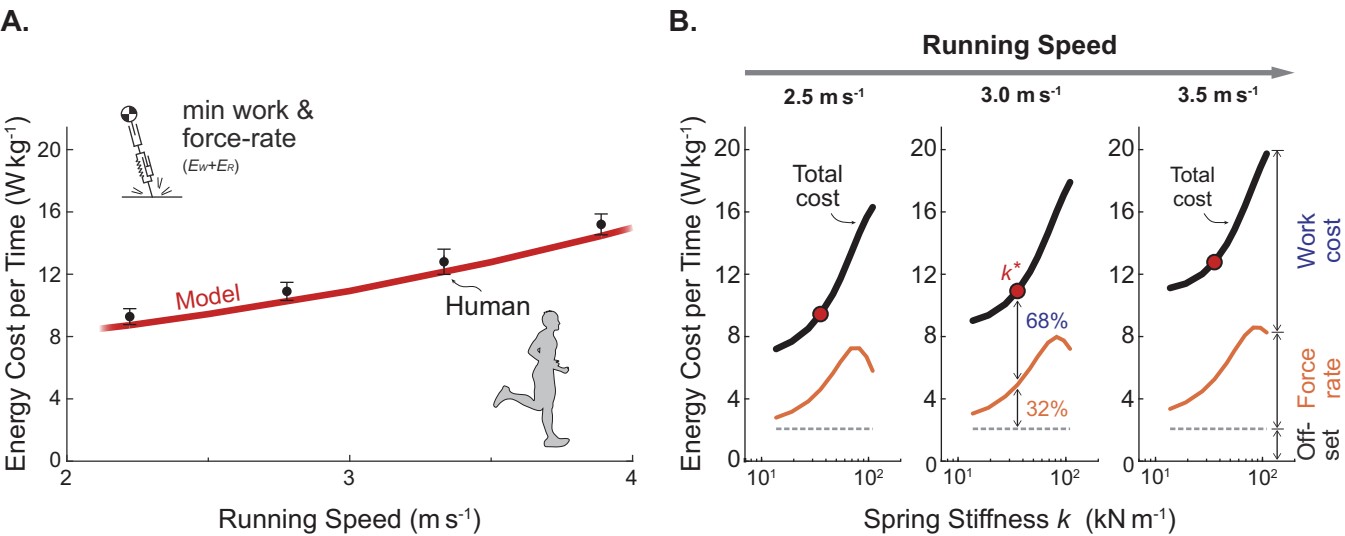

**Fig 8. Unified model energy cost vs. speed and spring stiffness, including force-rate cost.** (A) Energetic cost per time versus speed is shown for the Actuated Spring-mass model ($k^*$ of 35.6 kN m$^{-1}$, $\varepsilon^*$ of 0.5·10$^{-3}$; red curve) and for empirical metabolic data of human subjects running on a treadmill (mean ± standard deviation; [48]). Model cost includes costs for work and force-rate, plus a constant offset associated with human resting metabolism (dashed horizontal line). (B) The model's energetic cost is shown for three speeds $v$ (2.5–3.5 m s$^{-1}$) and with parameter variation of spring stiffnesses $k$ (13.7–109.5 kN m$^{-1}$), with total cost (black lines), force-rate cost (difference between offset and magenta lines), and actuator work cost (difference between total and magenta lines). The unified model's spring nominal stiffness $k^*$ is indicated (red line in A, red symbol in B).

presence of a force-rate cost makes particularly stiff springs more costly to the model and may indicate benefits of some compliance when running.

## Unified model explains energetic cost of running on inclines

The model may also be applied to uphill and downhill running. In humans, metabolic cost asymptotically converges toward the costs of muscle performing positive and negative work (about 25% and -120%, respectively; [11]). At intermediate slopes, the cost smoothly transitions between these two extremes, passing through the cost for level running. The unified model produces a similar cost curve (Fig 9A), with similar asymptotes. However, the force-rate cost adds to work costs in such a way that the human asymptotes are actually achieved with slightly different positive and negative actuator efficiencies (32% and -105%), though consistent with estimates on cross-bridge efficiency [49].

The model's energetic cost is dominated by positive and negative work at steeper upward and downward slopes, respectively (Fig 9B). Of course, increasing work is required of steeper slopes, but force-rate becomes less costly at those extremes. Additionally, for slopes surrounding zero, the force-rate cost contributes to the relatively smooth transition from positive to negative efficiency tangent lines identified by Margaria [11]. The minimum of the cost curve occurs approximately where passive energy dissipation approaches the net negative mechanical work of descending the ground slope (about -0.08 slope) and is consistent with simple collision models indicating optimal running slopes [50]. The force-rate cost is relatively high for shallow slopes and level ground, because it favors more impulsive forces that take advantage of passive dissipation to reduce active negative work. While even more passive dissipation at steeper negative slopes could reduce work costs further, this would come at a higher force-rate cost. In fact, it is less costly overall to actively dissipate energy at steeper slopes to avoid a high force-rate, but at increased work cost.

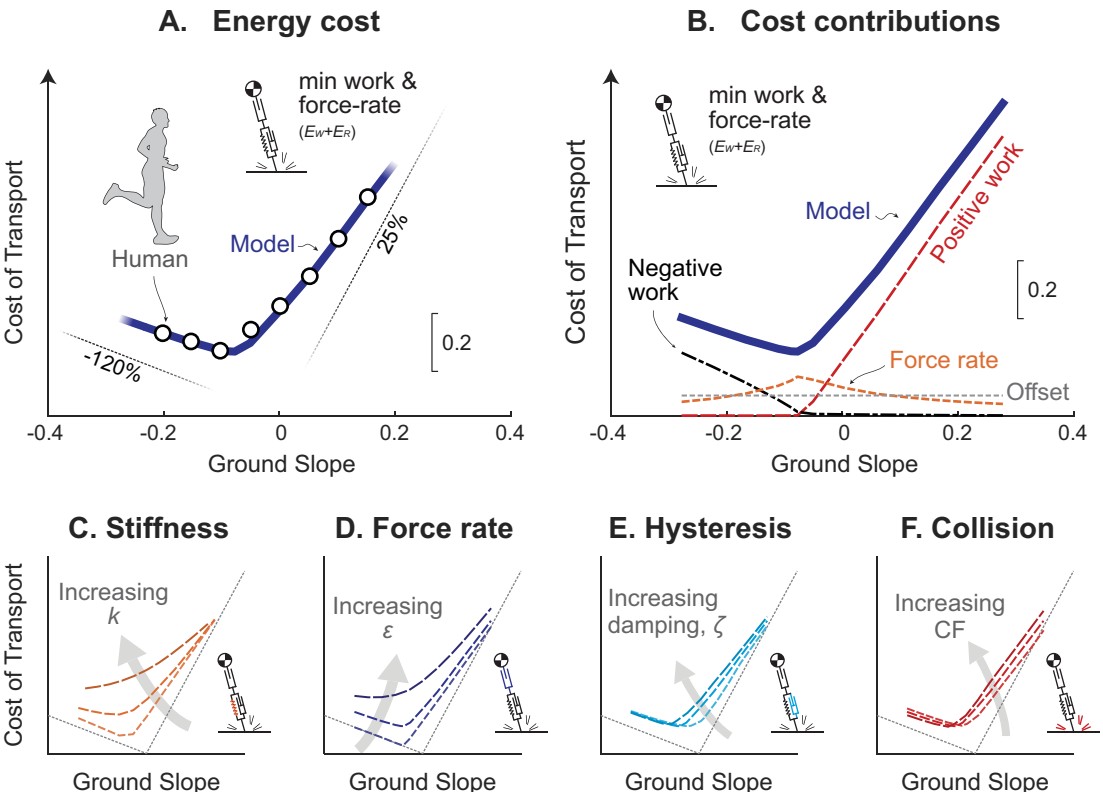

**Fig 9. Energetic cost of running vs. ground slope for unified Actuated Spring-mass model.** (A) Model Cost of Transport (solid line) compared to humans (circles; [11]). Also shown are asymptotes (thin lines) for muscle efficiency of positive and negative mechanical work (25% and -120%, respectively). (B) Contributors to model Cost of Transport: positive work cost, negative work cost, force-rate cost $E_R$, and a constant offset. Parameter sensitivities are included for varying (C) stiffness $k$, (D) force-rate coefficient $\varepsilon$, (E) hysteresis (damping ratio $\zeta$), and (F) collision fraction CF. Each trace indicates variation from lowest to highest parameter values: $k$ ranging 13.7 kN m$^{-1}$ $-\infty$, $\varepsilon$ ranging 0–2·10$^{-3}$, $\zeta$ ranging 0–0.2, CF ranging 0–0.06. All model results are for nominal running at speed of 3 m s$^{-1}$ and step frequency of 2.94 Hz.

Parameter variation was used to assess the model's cost sensitivity to spring stiffness $k$ (Fig 9C), force-rate coefficient $\varepsilon$ (Fig 9D), damping ratio $\zeta$ (Fig 9E) and collision fraction CF (Fig 9F). Costs generally increase with each of these parameters, particularly for shallow and level slopes. For example, increasing spring stiffness resulted in greater energy cost, largely due to increased force-rate cost associated with more impulsive forces. Greater force-rate coefficient was also more costly, since total cost is proportional to the coefficient (Fig 6). Increasing either dissipation parameter resulted in relatively modest increases in cost, mainly because more dissipation must be offset by more active work. But the overall result is that, even for extreme parameter variations, the cost of running up or down steeper slopes still tends to asymptote toward positive and negative work efficiencies. On shallow and level slopes, each parameter contributed toward a non-zero cost, resulting in an overall cost comparable to human (Fig 9A).

## Discussion

We have proposed several additions to the Spring-mass model that help to explain the energy expenditure of running. The Actuated Spring-mass model includes passive energy dissipation, active work, and an additional energetic cost related to force-rate. In combination, these elements show how running can still cost substantial energy, even though series elasticity acts

conservatively to reduce the active work required of muscle. We next re-examine each of the elements individually, to consider both the justification and the contribution of each to an overall model of running.

## Active work dominates energy expenditure despite elastic return

The primary energetic cost observed in the models considered here was for active work. Empirical estimates based on work during human running, along with assumed 50% elastic energy return, suggest that active work could account for 76% of the energetic cost [21]. Our model also had substantial elastic energy return, for example 53% at a speed of 3.9 m s$^{-1}$ (Fig 7), yet mechanical work still accounted for 66–70% of the overall energetic cost over the range of speeds considered.

The unified model actively performed both positive and negative work (Fig 7). Empirical measurements reveal modest active lengthening (and thus negative power) during early stance in humans (vastus lateralis; [47]) and in turkeys (lateral gastrocnemius; [5]). The perspective provided by our model is that such active negative work should generally be avoided if work were the only energetic cost. But active dissipation may be justified by opposing costs such as for force rate (Fig 8), which justify performing active negative work and extending stance durations (e.g. Fig 7C), but at the cost of yet more positive work to offset the active dissipation. The performance of active positive and negative work on level ground also explains the smooth transition in cost between uphill and downhill slopes. A leg that actively performs negative work on the level should simply perform more such dissipation for downward slopes, and less for upward slopes, and similarly for positive work, with cost asymptotes defined by work performed against gravity alone [11]. Even if passive elasticity performs most of the work of running, the remaining active work by muscles [16,17,47] could still explain much of the overall energetic cost.

The model also reproduces empirical correlations with energy cost. Kram and Taylor [51] observed energy costs increasing with speed, and proportional to body weight divided by ground contact time. The present unified model also yields similar correlations (cost proportional to inverse contact time, and inverse contact time to speed, $R^2 = 0.95$ and $R^2 = 0.98$ respectively), but as an outcome of optimizing costs for work and force rate. In fact, a simple analysis demonstrates that mechanical work of a series actuator can explain this proportionality explicitly (in supplementary material of [21]). We do not consider contact time to be a direct determinant of cost, because running (with constant body weight) with an especially flat CoM trajectory results in both greater contact time and greater energy cost [52]. Rather, a mechanistic energy cost from actuator work and force can potentially explain why quantities such as contact time can appear correlated (or not) with cost.

## Passive dissipation is a major determinant of mechanics and energetics of running

There are several features of running that are reproduced by the inclusion of dissipation. Dissipative losses are the primary driver of active mechanical work, which is needed to offset losses and obtain the zero net work of a periodic gait cycle. Even a relatively small amount of dissipation can be costly. For example, the unified model passively dissipated less than 5% of the body's mechanical energy and passively returned 59% of positive shortening work at 3 m s$^{-1}$ (compared to 60% in turkey; [8]), yet the remaining active positive work can still explain 61% of the net metabolic cost of equivalent human running. We also found dissipation (particularly collision loss) to increase substantially with running speed, and therefore contribute to greater energy expenditure rate (Fig 8).

Passive dissipation also helps to explain some time-asymmetries in running. For example, in humans the vertical ground reaction force increases faster early in stance than it decreases later in stance (Fig 7B). Such asymmetries have previously been attributed to the muscle force-velocity relationship [53] and to dissipation [54]. The present model predicts such asymmetry to be energetically optimal (Fig 7B). Given the leg extends after mid-stance when forces are already decreasing, the leg must undergo net extension by the time of take-off (Fig 7D). The model also predicts a more vertical leg orientation during touchdown versus at take-off to reduce dissipation similar to humans [54]. Other models have also demonstrated the economy of asymmetrical trajectories in bird running [30] but do not include a brief transient in ground reaction force and work at touchdown (Fig 7B and 7C), observed prominently in human foot contact.

Asymmetrical trajectories are not observed in the models without passive dissipation. The Spring-mass model produces more time-symmetric trajectories (Fig 3) lacking initial transients and predicts zero active work. The Actuator-only model also produces symmetric trajectories since it has no passive dissipation. However, it lacks passive elasticity resulting in cost over twice that of equivalent human running (Fig 3, bottom column). The time asymmetries in force and work profiles might seem like minor details, but here they are emergent from energy optimality. Passive dissipation helps to explain these asymmetries and is a major contribution to the energetic cost modeled here.

## Elastic tendon is critical to energy economy but not the pseudo-elastic mechanics of running

We next re-examine whether long, compliant tendons are helpful for locomotion economy [5,6,55]. If the total work per step from muscle and tendon were fixed, a more compliant tendon could indeed perform more of the work passively, and thus reduce the energetic cost. But in running, the total work need not be fixed, and changing the series stiffness also yields a different optimal trajectory (Fig 4), and a different amount of total work for a given speed and step length. In our model with zero force rate cost coefficient $\varepsilon$, a stiffer spring is actually more economical at higher speeds (3.5 m s$^{-1}$, Fig 5). The optimal trajectory yields lower spring displacement and hysteresis loss, and thus, less active work.

Another potential factor for compliance is the proposed force-rate cost. Avoidance of that cost can favor more active work (Fig 8), and overall cost may indeed be reduced with more compliance. More complex running models have also suggested that the optimal compliance may actually be different for each muscle [56]. Of course, there may also be other benefits to tendon compliance beyond economy [57]. But for running economy alone, there is no general prescription that favors greater tendon length or compliance.

Tendon compliance has long been implicated in the spring-like mechanics of running. This is manifested in CoM trajectories, ground reaction forces, power, and vertical work loop curves (Fig 3), which are all suggestive of elastic behavior. But all the models considered here, including those with dissipation and actuation, and even those with no elasticity at all (Actuator-only Model, Fig 3) also exhibit similar pseudo-elastic behavior. In models, springs are also not critical to the economical advantage of running over walking at high speeds [14], nor to the general cost trends for running on slopes (Fig 9C). This is not to diminish the importance of elasticity, which allows leg muscles to operate at lower and more efficient shortening velocities [5], and store and return substantial energy. But the basic resemblance of running to elastic bouncing, and the associated pseudo-elastic mechanics (e.g., Fig 3), should not be regarded as evidence of true elasticity in running.

### Force-rate cost contributes to mechanics and energetics of running

In addition to the cost of active work, our model also includes an energetic cost for force rate. Such a cost has been experimentally observed in tasks such as cyclic muscle contractions [24,35], and was included as an energetic penalty for rapid transients in force production. We found that work as the only energetic cost tended to favor overly impulsive running motions (Fig 6). The force-rate cost acts as a trade-off against work, resulting in reduced peak ground reaction forces and longer stance durations, more similar to human data (Fig 7). The trade-off also makes it economical to perform active negative work, and to favor more compliant series elasticity, contributing to more human-like mechanics (Fig 7) and energetics (Fig 8). Work alone also reveals a particularly economical strategy for running downhill, where only a minimal amount of active negative work is performed to dissipate gravitational potential energy. However, humans expend more than the predicted amount of energy, perhaps because the force-rate makes such a strategy more costly, resulting in a smoother transition toward the negative work asymptote observed experimentally [11]. The force-rate cost produces these effects despite a relatively modest energetic cost. In the unified model, force-rate accounts for only 32% of energetic cost, compared to 68% for work (Fig 8). And across slopes, work alone predicts too low an energetic cost for running (Fig 9D). A cost such as force-rate is thus helpful for explaining human-like energetics.

### Implications for legged robots

The present study may be relevant to running robots that employ spring-like behavior. In the SLIP (spring-loaded inverted pendulum) paradigm, a controller causes the overall leg to behave like a spring, despite internal dissipation. Robotic dissipation includes actuator heat and transmission losses (analogous to actuator work efficiency and hysteresis in our model) as well as from interactions with the environment (e.g. collisions) [58]. With SLIP control, the stiffness is sometimes selected to resemble animal gait [59], just as our model resembles human. Our results suggest that SLIP may actually be reasonably economical, because our model, similar to others modeling dissipation with more detail [27,28], yield optimal force profiles that are still remarkably spring-like. However, closer examination also reveals that better economy is achieved with small but significant differences such as force asymmetry (Fig 7). We also note that matching a gait to human or animal is not necessarily the best option for economy, which might favor a quite different stiffness (Fig 8B). There are also potential benefits to including passive dissipation, which can yield improved stability [29] and velocity estimation, which is considered important for robust control [60,61]. Robotic controllers can take advantage of passive dissipation by modeling it explicitly (e.g., [62]) and will respond differently when optimized for economy.

### Limitations

The running model proposed here has a number of limitations, with regard to passive dissipation, actuator costs and running dynamics. For dissipation, we modeled hysteresis during leg compression alone to reproduce empirically estimated energy losses, without a detailed model of hysteresis mechanics. Similarly, we modeled collision losses with a simple reduction in momentum at touchdown, without considering the direction and mechanical properties of the body mass experiencing impact. To our knowledge, most models of running do not include explicit dissipation. We consider the present model to be an indicator that dissipation is important, but also in need of better-informed dissipation mechanics. The same is true for our model of energetic cost, intended to extend Spring-mass models with a highly simplified dependence on active mechanical work. Our model also stands to be improved with other

features potentially relevant to running, such as muscle moment arms and force-velocity relationships (e.g., [63]).

There are also limitations of the proposed energetic cost for force rate, which is not included in most other running models. This cost is intermediate in complexity between abstract "effort" costs such as force squared [64,65], and more detailed models of muscle energetics [66,67], while also being supported by empirical data (e.g. [23–25,34,35]). But the specific dynamics of such a cost in muscle are not well understood, with considerable uncertainty in the appropriate formulation. For example, we have found different derivative order and exponent (e.g., 2 and 1, respectively for Eq 3) to yield similar forces [26] and agree well with empirical energy costs [23]. We therefore regard the force-rate cost as a provisional implementation that stands for considerable improvement.

Similar to the Spring-mass model, the present model is also a gross simplification of human body dynamics. It is intended only to model basic features of running, such as pseudo-elastic mechanics and overall energy expenditure. More complexity would be required to address motion of multiple-jointed models, intersegmental dynamics, and activation or co-contraction of multiple muscles. For example, the model neglects a swing leg, whose active motion may also cost energy [22], and be exploited to modulate collision losses via active leg retraction just before touchdown [68,69]. We also fixed step length and frequency with respect to running speed, whereas these could also be included in optimization for preferred gait parameters (e.g., [38]). The present model only provides basic suggestions, that dissipation and energetic costs for work and force rate may be important for running. These suggestions are intended to apply to more complex models of running, but this remains to be tested.

## Conclusions

The energetic cost of running is not addressed by the classic Spring-mass model of running. Although elastic tissues store and return energy during stance, there is still some dissipation due to touchdown collision and hysteresis. For steady gait, these losses must be restored by active, positive work from muscle. The present Actuated Spring-mass model shows that even a relatively small amount of work can still incur a substantial energetic cost. It is also particularly costly to perform active negative work, because the associated losses must be restored by additional positive work. Muscles may also expend energy for high rates of force development that make it economical to perform some active negative work, ultimately helping to explain the energetics of running at different speeds and slopes. Spring-like forces and other mechanics emerge from an actively controlled model optimized for economy, even if there were no elasticity. Series elasticity may be critical to saving energy, but active work and passive dissipation appear important for determining the energetic cost of running.

## Supporting information

**S1 Text. Dynamic optimization model details.**
(DOCX)

## Acknowledgments

The authors would like to thank Arash Khassetarash (University of Calgary) for sharing experimental data.

## Author Contributions

**Conceptualization:** Ryan T. Schroeder, Arthur D. Kuo.

**Data curation:** Ryan T. Schroeder.

**Formal analysis:** Ryan T. Schroeder.

**Funding acquisition:** Arthur D. Kuo.

**Investigation:** Ryan T. Schroeder, Arthur D. Kuo.

**Methodology:** Ryan T. Schroeder, Arthur D. Kuo.

**Project administration:** Ryan T. Schroeder, Arthur D. Kuo.

**Resources:** Arthur D. Kuo.

**Software:** Ryan T. Schroeder.

**Supervision:** Arthur D. Kuo.

**Validation:** Ryan T. Schroeder, Arthur D. Kuo.

**Visualization:** Ryan T. Schroeder, Arthur D. Kuo.

**Writing – original draft:** Ryan T. Schroeder, Arthur D. Kuo.

**Writing – review & editing:** Ryan T. Schroeder, Arthur D. Kuo.

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
