## [Decision Letter · Decision Letter 0]

22 Jul 2021

Dear Dr. Schroeder,

Thank you very much for submitting your manuscript "Elastic energy savings and active energy cost in a simple model of running" for consideration at PLOS Computational Biology. As with all papers reviewed by the journal, your manuscript was reviewed by members of the editorial board and by several independent reviewers. The reviewers appreciated the attention to an important topic. Based on the reviews, we are likely to accept this manuscript for publication, providing that you modify the manuscript according to the review recommendations.

In particular, it is essential to take note of the recommendation of both reviewers that citation and discussion of a wider range of recent relevant papers is needed, to set your work in context.

Sincerely,

Barbara Webb

Associate Editor

PLOS Computational Biology

Daniel Beard

Deputy Editor

PLOS Computational Biology

[LINK]

Reviewer's Responses to Questions

**Comments to the Authors:**

Reviewer #1: Summary:

The authors offer an improved computational model of the energetic cost of running in humans by accounting for aspects of running that are generally ignored in models of legged locomotion. These aspects include losses in the virtual leg springs, losses due to collision with the ground, the rate at which the leg musculature/actuator produces work, and the cost of negative work. It is common sense that the two losses might be important and the authors cite sufficient literature to argue that the rate of work is important. By taking these additional costs into consideration, the authors show a cost breakdown that corresponds roughly with predictions of the percent of energetic cost from active muscle work from the literature (assuming reasonable elastic energy return from the tendons). They are also able to extend their model to capture the changes to energetic cost of a human running up and down slopes. Overall, this paper is a nice contribution to the literature on dynamic legged locomotion.

Major comments:

I am concerned that the flat ground human running data all comes from one individual. The paper that this single individual's data was drawn from had 10 young male runners -- is there a reason that only one runner's data was selected for comparison to the model?

In this paper, the spring-actuator model has the spring (with damping) on top next to the center of mass and the actuator lower down in the leg. Intuitively, I expect to see the actuator closer to the center of mass and the damped spring closer to the toe. If the actuator is a rough analog for the muscles of the leg and the damped spring is a rough analog for the tendons, then actuator-on-top seems to be the more common body plan for mammals and birds. Also, having the damped spring being close to the toe would allow it to contribute to rejecting perturbations from the ground and take up energy during collisions -- legged robots with series elastic actuators tend to put the springs nearer to the end effectors for this reason. A sentence or two explaining why the authors made this modeling choice would be very helpful to me as a reader.

Finally, I want to point out an option which is available to the authors. Since they have a comparison between the actuator-only and spring-actuator models, they could comment on the differences between biological and robotic legged locomotors. Actuator-only locomotion schemes are common for robots with legs, and some discussion of when actuator-only legged robots can be used to test models of locomotion in the physical world would increase the audience for the paper. There are also bipedal robots like Cassie which have springs in their lower legs (whereas the model in this paper puts the springs closer to the center of mass), and the authors could comment on the choice of spring placement. I understand that relating the results to legged robots is outside the current scope of the paper and the authors should not interpret this suggestion as a requirement -- merely a suggestion if they are interested in broadening their audience. Anecdotally, I have found that researchers working with legged robots have a lot of interest in how to judiciously use damping to improve locomotion performance.

Minor comments:

Why is "Spring-mass model" capitalized in this way? I expect either "spring-mass model" or, if the authors want to refer to the model using an acronym, "Spring-Mass Model (SMM)". Similarly, I suggest changing "Actuator-only" to "actuator-only" and "Actuated Spring-mass" to "actuated spring-mass".

29: Even if the tendons are very good springs, it would be impossible for them to be totally lossless. We would expect them to dissipate SOME energy, requiring the muscles to add that energy back in order for the animal to maintain a steady-state run. I would suggest taking out the phrase "and why the muscles do any work at all". If the authors wish to make the point that their model accounts for dissipation in the tendons where other models do not, I think it is important that they clarify this. Before reading the paper, it was not obvious to me whether the classical model the authors refer to assumes lossless springs or not.

61: Flight  flight

Since the authors make the point that models of legged locomotion rarely consider damping, it seems important to include some other recent work on the importance of including damping in computational models. I am personally interested in how the authors would relate their work to this article which came out in August 2020, but the authors may find papers that are a better fit (I realize that this paper is talking about economy of motion rather than stability):

Heim S, Millard M, Le Mouel C, Badri-Spröwitz A. 2020 A little damping goes a long way: a simulation study of how damping influences task-level stability in running. Biol. Lett. 16: 20200467. http://dx.doi.org/10.1098/rsbl.2020.0467

Reviewer #2: The manuscript “Elastic energy savings and active energy cost in a simple 2 model of running” reports on the results of a simulation/optimization study on a conceptual running model that investigates different potential contributions to energy expenditure. The model is a simple point mass with a mass-less leg that features a compliant element with parallel damping and series actuation. Energy optimal motions are found through numerical optimization for which the cost function is composed of a work term (integral of weighted positive and negative mechanical power) and a force-rate term (integral of force-rate squared). The model is most similar to the one presented in [1] for which energy-optimal motions have also been identified. The main difference with regard to the model is the inclusion of (virtual) collision losses. I would consider the main contribution of the paper to be the proposed consideration of a force-rate cost term, which is carefully studied. In particular, this force-rate term might stipulate a pseudo-elastic behavior (with active negative work) that could explain the apparent elastic characteristics of running while also explaining the associated energy cost. Without being able to provide conclusive evidence that this is indeed what happens in biological systems, the results suggest that some type of force-rate term (not necessarily exactly as the one presented) might explain characteristics observed in human running and they clearly establish a sound alternative explanation to a purely elastic behavior. These conclusion are formulated with the appropriate care, and limitations of the used model and methods are adequately addressed.

Overall, the paper is very well written and nicely framed. My comments are mostly positive, yet I still see some potential for improvements. Most importantly the inclusion of a discussion of currently unreported prior work. With regard to the presentation, the explanations in the manuscript sometimes require the reader to infer some intermediate steps themselves. This could be clarified to help with the readability of the paper.

• While the paper is well grounded in the existing literature, it seems to be missing some recent work out of the collaborations of Monika Daley and Jonathan Hurst (e.g., [1]), which studies work-minimal motions of running birds using a nearly identical model with damping losses and active (series elastic) inputs. I feel that these contributions need to be discussed in the context of your proposed model (L101).

• With simple models as the one presented in this paper, a particular important question is the choice of model parameters. These are well justified in the manuscript, but I would suggest to summarize them in a table for a better overview.

• What puzzled me initially was the different physical nature of the actuators in Fig 2. As the actuator in 2b acts against a mass M, I immediately envisioned it as a force-source, whereas the actuator in 2c appears to be a kinematic constraint (which is not acting against a mass). Furthermore, it was not immediately clear how the force-rate would be computed for case 2b. These apparent differences are cleverly resolved by using the jerk of the leg length as input, yet that information is hidden in the supplementary information. I would suggest to expand a little bit on this issue after line 181 and explicitly state that jerk is the input to all actuators (and force-rate a dependent quantity that follows from this kinematic constraint).

• While the paper is very mathematical in its methods, the description of the different scenarios relies often purely on verbal descriptions. It might help for a more mathematically focused audience to include the relevant symbols whenever possible. (e.g., L179: “Model states included the position (x,y) and velocity (\\dot{x}, \\dot{y}) of the point mass body and the first and second derivatives of the actuator’s length (\\dot{l} and \\ddot{l}) to …”; L192: “tendon stiffness k … and force-rate coefficient \\epsilon” and in many other places.)

• L154: a) Shouldn’t this read “L_m(t)”? b) Wouldn’t it be more precise to state that the jerk of L_m(t) was the control input. I know that technically defining L_m(t) defines the jerk, but there are some implicit assumptions in play that, for example, require L_m(t) to be three times differentiable for a trajectory to be physically realizable and to have a well-defined finite work-rate cost.

• Eq 3 uses the lunate epsilon symbol, whereas elsewhere in the document the Greek lowercase epsilon is used.

• L172/L193/L246/L275: From a philosophical point of view, the nature of \\epsilon could be two-fold. One can either think of it as a relative weight in a weighted cost function (in which case it would be placed better in eq. 1 and not in eq.3) or as a biomechanical parameter that has a clearly defined unit and needs to be identified (and for which we don’t know a-priori if it is unequal to 0). I believe the paper only wants to do the latter, yet sometimes falls into wording that would be more appropriate for the former (such as: “With work as the sole cost”). I would a) suggest to be a bit more precise in the wording (“for a cost coefficient \\epsilon=0”) b) to define the range to be from 0 to 0.02) , and c) to immediately clarify the units and nature of \\epsilon.

• Along those same lines, I would suggest to generally avoid expressions such as “with *work* as the sole cost,…”, as it lacks a bit precision. I was first thinking of a completely different cost function of mechanical work or positive mechanical work, before realizing that you meant E_W, which is not work, but the “cost of work”. (I know this is getting very detailed, but the paper is at a point where only very detailed comments will make it even better).

• L328: To clarify it might help to state that the work is needed to compress the spring to increase the force. I was first puzzled since force does not show up in any cost term.

• L365: Your reporting of the efficiency constants here reminded me that these are not reported in the first part of the results. Their particular choice will likely not influence the results as long as negative work cannot be recovered, but their choice influences the relative weight of \\epsilon, so they must be reported. Again, a table (maybe in combination with the one requested above) would help with clarity.

• L364: how was \\epsilon and k determined? Are these parameters fitted to data or follow from optimality criteria? This seems very important, as a key message of the following results is similarity to human data.

• L373/L508: If I remember correctly, this shape is also reported in the above mentioned reference.

[1] “Don’t break a leg: running birds from quail to ostrich prioritise leg safety and economy on uneven terrain”, The Journal of Experimental Biology (2014) 217, 3786-3796

**Have the authors made all data and (if applicable) computational code underlying the findings in their manuscript fully available?**

Reviewer #1: Yes

Reviewer #2: Yes

PLOS authors have the option to publish the peer review history of their article (what does this mean?). If published, this will include your full peer review and any attached files.

Reviewer #1: **Yes: **Sonia Roberts

Reviewer #2: No

Figure Files:

Data Requirements:

Reproducibility:

References:

---

## [Decision Letter · Decision Letter 1]

2 Nov 2021

Dear Dr. Schroeder,

We are pleased to inform you that your manuscript 'Elastic energy savings and active energy cost in a simple model of running' has been provisionally accepted for publication in PLOS Computational Biology.

Best regards,

Barbara Webb

Associate Editor

PLOS Computational Biology

Daniel Beard

Deputy Editor

PLOS Computational Biology

Reviewer's Responses to Questions

**Comments to the Authors:**

Reviewer #1: You have addressed all of my points of concern and clarified elements of the original paper that I found confusing. I am happy with the revisions.

Reviewer #2: My prior comments have been addressed to my satisfaction.

**Have the authors made all data and (if applicable) computational code underlying the findings in their manuscript fully available?**

Reviewer #1: Yes

Reviewer #2: None

PLOS authors have the option to publish the peer review history of their article (what does this mean?). If published, this will include your full peer review and any attached files.

Reviewer #1: **Yes: **Sonia Roberts

Reviewer #2: No

---

## [Editor Report · Acceptance letter]

16 Nov 2021

PCOMPBIOL-D-21-00861R1 

Elastic energy savings and active energy cost in a simple model of running

Dear Dr Schroeder,

I am pleased to inform you that your manuscript has been formally accepted for publication in PLOS Computational Biology. Your manuscript is now with our production department and you will be notified of the publication date in due course.

With kind regards,

Zsofia Freund
